# Caught between relief and unease: How university students' well-being relates to their learning environment during the COVID-19 pandemic in the Netherlands

Lisa Kiltz◉*, Marjon Fokkens-Bruinsma, Ellen P. W. A. Jansen

Behavioural and Social Sciences Faculty, Department Teacher Education, University of Groningen, Groningen, The Netherlands

* l.kiltz@rug.nl

## Abstract

Previous literature shows that university students are particularly vulnerable to psychological ill-being. Also throughout the COVID-19 pandemic, stressors ranging from uncertainty to disruption of social lives have influenced their well-being. Resilience as a psychological resource could help students deal with such crises. Furthermore, students' learning environment can substantially determine their well-being and resilience, by satisfying their basic psychological needs for autonomy, competence, and relatedness. The present study aims to longitudinally investigate students' well-being and resilience in relation to their learning environment. To this end, we interviewed six participants, of which two were university students, two university teachers, one study advisor, and one student psychologist. With a longitudinal interview study with four dates of measurement, spanning the pre to mid-COVID-19 pandemic period, we gathered commentary about the evolution of student well-being, resilience factors, and the effects of the learning environment. To analyse the interview data, we used thematic inductive and deductive coding. The participants confirmed the postulated stressors, but also positive consequences for student well-being, including resilience growth. Interviewees also reported a variety of resilience factors, both within the individual (e.g. social support) and within academia (e.g., impaired student-teacher relationship, diminished sense of belonging). Furthermore, the interview data indicate changes in teaching related to students' needs for autonomy, competence, and relatedness, which in turn have consequences for learning and engagement, including challenges, opportunities, and positive outcomes. These findings, connecting the learning environment to student well-being and resilience, may help reshape academic systems for the post-pandemic future.

**Data Availability Statement:** As indicated, the raw transcripts are too sensitive to share openly. They

concern participants' well-being, which is considered sensitive information, and nearly impossible to fully anonymise as a lot of in-between information may be indicative of a specific person. This decision was taken together with the ethics committee of the department of Teacher Education of the Faculty of Behavioural and Social Sciences at the University of Groningen, which can be contacted at ec-bss@rug.nl

**Funding:** The author(s) received no specific funding for this work.

**Competing interests:** The authors have declared that no competing interests exist.

# Introduction

*First, it was a relief and then it was a bit more stress because it was such uncertainty about many things and then . . . things became a bit more certain and more things became online, so they [took place], but online. So I think that helps, too.*

—University student (April 2020)

At the beginning of 2020, the COVID-19 pandemic hit the world abruptly and with a societal impact rarely seen in modern history. Considering that the "psychological footprint" of COVID-19 may be even greater than its "medical footprint", [1, p712] it is critical to investigate its psychological effects. For instance, governments introduced persisting restrictions on social and public life that have substantially reshaped the academic learning environment (LE). For the scope of the current study, we define the LE as an environment in which stakeholders act (e.g., students, teachers, and support staff), and in which educational (e.g., teaching modes) and structural aspects (e.g., support systems) play crucial roles. As university students, hence, faced unexpected remote teaching, social distancing, and lockdown within their LE, their well-being and resilience resources may have been impacted. Therefore, we investigate academic well-being and resilience in relation to students' LE shortly before and throughout the initial months of the COVID-19 pandemic.

## Student well-being in pandemic times

With the substantial effect of societal restrictions on people's lives, the COVID-19 pandemic has affected overall well-being, including social, emotional, physical, and psychological facets. [e.g. 2; for a thorough definition of student well-being, see 3] Taylor [4] identifies various pandemic-related stressors associated with well-being, including uncertainty, disruption of routines, separation from family, social isolation, financial insecurity, and closure of educational institutions. Additionally, Holmes et al. [5] propose a general sense of loss as a psychological consequence.

Young adults have been particularly affected by the current pandemic [6–8]. For instance, studies from the UK, US, and Poland implemented shortly before and into the pandemic found a significant increase in psychological distress and ill-being among young adults [9–12]. Particularly students' experienced impaired psychological well-being compared with other populations [2,13]. They reported elevated symptoms of depression and anxiety as well as decreased quality of life [14–17]. Likewise, a longitudinal research project in the Netherlands found that students report the absence of social interaction and connection, accompanied by higher stress [18]. In summary, prior research consistently highlights students' compromised well-being throughout the COVID-19 pandemic. Individual and institutional resources, however, may counteract this tendency and support student well-being.

## Students' resilience in pandemic times

Resilience refers to an individual capacity to bounce back from a stressor and regain one's former psychological health [19,20]. Studies have repeatedly demonstrated that resilience promotes student well-being and counteracts negative experiences [21–23]. In times of extraordinary strain, such as the pandemic, resilience may be the key to handling stressors and minimising their impact [24]. However, not everyone possesses such resources, leading to individual differences [25,26]. For instance, more resilient students perceived less stress during COVID-19 [7]. Likewise, students' resilience moderated pandemic-related stressful

experiences and acute stress disorder [27]. Also in the general population, more resilient individuals experienced fewer worries, anxiety, or depressive symptoms [28].

When bouncing back from adversity, individuals mobilise personal and environmental resilience resources [29]. Several studies aim to categorise such resources, such as Mansfield et al., [30] who classify four overarching categories: personal and contextual resources, strategies, and outcomes. Also regarding the COVID-19 pandemic, Chen and Bonanno [25] state that resilience depends on both individual and structural characteristics, including family and community surroundings. Nonetheless, such resilience and risk factors require further investigation. Research investigating well-being specifically for university students has already identified several protective factors: enhanced social support, shared emotions, time for family and hobbies, self-efficacy, and effective coping [27,31,32]. However, most research thus far does not focus on resilience as essential to support student well-being in times of crisis. Our study aims to further define and categorise factors that could promote students' resilience.

Beyond its cross-sectional definition, the resilience concept implies that experiencing adversities may result in a process of positive adaption throughout time, the so-called resilience growth [19,33]. Within this process, more severe stressors might produce greater resilience than less intense stressors [34]. Therefore, people who have never experienced difficulties may be less likely to have already developed substantial resilience [19]. Students–who have less life experience than older people–may possess fewer personal resources [2,6,23]. For instance, Forycka et al. [15] found that more than two third of students reported low resilience throughout the pandemic. However, they might learn quickly how to cope with adversities and grow personally. Therefore, we take a longitudinal perspective on resilience development in students in the present study, as called for previously [25].

## Learning environment and basic psychological needs

From a systemic perspective, based on social constructivism [35,36], people are influenced by the system surrounding them. The academic system, for instance, is shaped by interactions between stakeholders, including students, teachers, and support staff. These interconnected stakeholders collectively determine both teaching and learning [37], emphasising the LE's nature as an interdependent system [38,39]. Consequently, the responsibility for students' well-being cannot solely be ascribed to the students; rather, a lack of well-being is a warning signal of a malfunctioning system [40,41]. Therefore, the LE's systemic factors are fundamental to investigating student well-being.

Within the LE, student well-being relies on the satisfaction of their basic psychological needs (BPN). These needs stem from the self-determination theory, identifying aspects that foster well-being [42] and have been associated with well-being also during the COVID-19 pandemic [43,44]. Educational structures can promote the BPN: autonomy, defined as a sense of freedom and control; competence, or self-efficacy and a capacity to thrive; and relatedness, defined as social support and sense of belonging [45,46]. Likewise, satisfying students' BPN through student-teacher relationships can lead to enhanced well-being and academic learning. In particular, autonomy can be stimulated by providing the freedom to shape one's studies. Competence relates to giving constructive feedback, and relatedness stems from investing in a respectful, appreciative relationship [47].

Because distance learning challenges social interactions, satisfying BPN in a disconnected environment can be undermined [47]. During the COVID-19 pandemic, the LE had to be adjusted abruptly, resulting in hastily realised and often insufficiently elaborated remote teaching [48,49]. Such virtual or hybrid academic surroundings have likely affected students' well-being negatively. For instance, pandemic-related stressors complicate satisfying students' BPN

[43,50]. More specifically, although distance learning may enhance students' autonomy by offering independence [51], losing freedom and control due to external restrictions could relate to its deprivation. Likewise, missing feedback and educational opportunities might result in fewer experiences of competence. Lastly, social distancing may deprive students of feeling relatedness or community [43,50]. These aspects may add up to a lack of need satisfaction: Many students have experienced decreased motivation and engagement (corresponding to a lack of autonomy), unproductiveness, mental overload (corresponding to a lack of competence), as well as isolation (corresponding to a lack of relatedness) [18,52]. Students' relationships with peers and teachers have especially suffered [53–55]. At the same time, interventions designed to foster BPN satisfaction during the pandemic promoted well-being [44,56]. Consequently, we propose that the pandemic has explicitly affected students' BPN within the LE and, in turn, their well-being.

### Research aims

Building on this framework, the present study focuses on three aspects: the pandemic's effects on student well-being, relevant resilience factors, and the role of the LE in this regard. Accordingly, we formulate three research questions:

RQ.1 How do university students, teachers, and support staff perceive student well-being and pandemic-related stressors before and throughout the COVID-19 crisis?

RQ.2 According to university students, teachers, and support staff, which resilience factors support students' well-being, how have they changed, and which factors offer the potential for resilience growth due to the COVID-19 crisis?

RQ.3 How do university students, teachers, and support staff perceive changes in the LE, relating to student well-being and their need satisfaction throughout the COVID-19 crisis?

## Materials and methods

With a longitudinal, semi-structured interview study, we aim to shed light on how the pandemic has affected students' well-being and resilience throughout the pandemic. We examine the effects systemically by including various stakeholders' perspectives—namely, university students, teachers, and other faculty members. Planned in pre-pandemic times, we had designed the study as cross-sectional; however, we decided to prolong it to span four measurement waves when the pandemic hit. The corresponding initial pre-registration can be found at https://osf.io/vu9ip; the adapted pre-registration at https://osf.io/uv2ze. As the situation changed quickly during these months, it appears essential to understand the situational context (see Fig 1). We recruited the participants at the end of 2019 and interviewed them for the first time before the pandemic hit the Netherlands. The second time of data collection then fell in the period of the first lockdown and the end of the academic year, with social distancing measures and the closure of universities in place. The third interviews, however, took place in the summer holidays, during which only social restrictions had remained, although remaining socially restricted. Lastly, t4 was similarly affected as t2 by the second lockdown in the Netherlands whilst the new academic year had just begun.

### Participants

Six interviewees participated in the study: two university students (one bachelor's and one master's student), two teachers, one study advisor, and one student psychologist. The

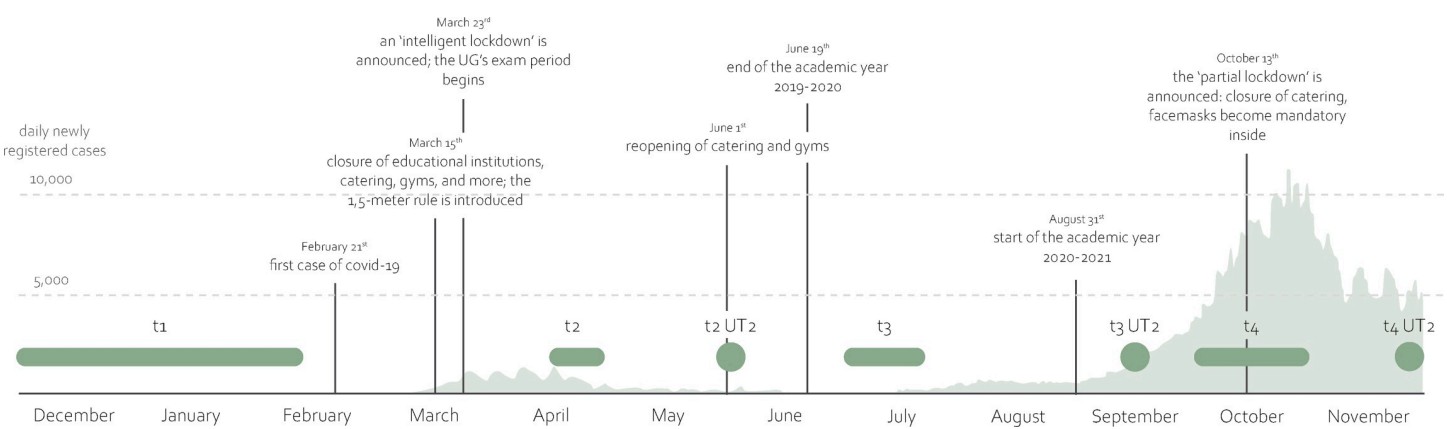

**Fig 1. The study's timeline, including the times of measurement and the relevant events surrounding the pandemic.** *Note*. After t1, one university teacher (UT2) was interviewed according to another time schedule and displayed as such. All societal events and pandemic developments concern the Netherlands. All university events concern the University of Groningen (UG). Sources: containmentnu.nl, coronavirus.nl, nos.nl, rijksonverheid.nl, rivm.nl, ukrant.nl, who.int.

participants were recruited personally and had already been in contact with the researchers because of their interest in student well-being. The students, both female, were 22 and 23 years old at t1; one was Dutch and the other an international student (within Europe). Both were in their final year at t1 and began new studies before t4 at new universities, one in the Netherlands and the other in her home country. The faculty members' ages ranged between 36 and 57 at t1, 50% were women, and their work experience ranged from 5 to 23 years. Participants studied or worked for various faculties, including arts, behavioural and social, medical, and spatial sciences.

## Interview and procedure

The ethics committee of the department of Teacher Education at the University of Groningen approved the initial study and the subsequent adjustments (i.e., shifting to a longitudinal study focusing on pandemic-related well-being; under TED-1920-S004). All six interviewees consented to participate in the study before the initial data collection. We administered a short questionnaire to collect socio-demographic information before the first interview.

The interviewer followed a semi-structured guide, similarly structured for all four times. The interview first focused on student well-being, then on the LE's impact, and finally on resilience factors that positively influence student well-being (see Table 1 and S1 File for further detail; we do not provide the interviews, transcriptions, or diaries due to the sensitivity of these data). The participants were interviewed either in person (mainly at t1) or online (mainly at t2–t4). These interviews lasted 34 to 100 minutes and were audio or video-recorded, depending on the interview setting. All recordings were transcribed verbatim. During this procedure, all identifiable information was replaced with pseudonyms. The participants received a transcription summary, allowing them to report potential misunderstandings or withdraw their data from the study.

## Measures surrounding the interviews

During the first interview, participants categorised aspects promoting student well-being that they mentioned. The interviewer collected keywords on sticky notes to depict the potential resilience factors. Then participants arranged the sticky notes, added headlines, and created connections. The resulting categorisations served as a basis for subsequent interviews. At t2,

**Table 1. Summary of the interview scheme and additional measures across all four time points, including example questions.**

| Time of measurement | Interview topic | Example question | Additional measure |
|---|---|---|---|
| t1 | Student well-being | Can you tell me what well-being at the university means to you? | |
| | Learning environment | Would you describe for me a situation in which you have felt supported by your department or the university in general? | |
| | Resilience | What do you do when you are feeling stressed out? | Categorising the resilience factors mentioned throughout the interview |
| t2 | Student well-being | Is there anything positive you learned about yourself now in this crisis? | |
| | Learning environment | What would you need from the university in the following months? | |
| | Resilience | What helps you at the moment to maintain a positive state of well-being? | Indicating which resilience factors mentioned at t1 had become more/less relevant or changed in nature |
| t3 | Student well-being | If I asked your teacher how the Corona Crisis changed studying, what do you think they would say? | |
| | Learning environment | What does the contact with your professor look like at the moment? | |
| | Resilience | How has your social network changed these last months? | |
| t4 | Student well-being | How has student well-being changed compared to t1? | Keeping a diary about specific events between t3 and t4 |
| | Learning environment | How connected do you feel to your new fellow students? | |
| | Resilience | How do you implement your coping strategies nowadays? | Indicating the resilience factors that help to better cope with future stressors |

for example, participants stated which factors had become more or less essential or changed in nature due to the pandemic. During the last interview, participants reported the factors' potential for resilience growth.

The students kept a diary between t2 and t3 for event-based data collection, including one challenging situation per week. Being less intrusive and closer to relevant events, this method helps gather affective data between measurements [57]. The students sent the resulting input to the researchers as a basis for the third interview. However, only one student completed the diary; the other reported having been too stressed to find the time.

Five months after the final data collection, we conducted an online focus group with four participants (excluding one teacher and the student psychologist) to allow them to reflect on the results. Using Google Jamboard, the participants suggested practical implications based on the initial results pertaining to the LE's potential for student well-being. Then they commented on the student-teacher relationship during the pandemic and suggested potential solutions for the growing distance.

## Analysis

In the initial coding phase, we coded all data partly inductively and partly deductively using Atlas.ti©. For RQ.1, the pandemic-related stressors disruption of social life, uncertainty, financial insecurity, daily routines, and a general sense of loss served as deductive codes. For RQ.3, the deductive codes encompassed the BPN, autonomy, competence, and relatedness. These

codes were enriched with additional codes emerging from the data until saturation. For RQ.2, we approached the analysis differently: one researcher first used the categories that participants created during the first interview and clustered them according to their similarities, then matched all resilience factors mentioned during the categorisation task with the category clusters. Subsequently, three researchers grouped and relabelled these factors to represent overarching themes. As some themes resemble Mansfield et al.'s [30] categorisation of resilience factors, we adopted their terms for these cases.

Based on this first coding phase, we created a codebook for each research question. Three researchers discussed these codes in shared reflection sessions and created final overarching themes. This process reduced the number of codes to 8 to 15 codes for each research question. We adjusted the initial codebooks accordingly and used them as a basis for the second coding round. To ensure interrater reliability, two independent researchers executed this coding stage, one for the whole data set and the other for 8 of the 24 interviews. They discussed and resolved discrepancies, then addressed any open questions in an additional, shared reflection session with all researchers. This process produced the adjusted final codebook.

## Results

In the following, we will give an overview over the clusters we identified throughout the analysis, structured according to the three research questions. We differentiate between clusters–an overall theme of various codes–and sub-factors related to these clusters–more detailed themes we found within these clusters. Fig 2 depicts an overview of these clusters (displayed in bold) and sub-factors (displayed in regular font); the S3 File encompasses a description of all codes.

### RQ.1 student well-being in times of crisis

**Well-being.** For student well-being, we identified well-being itself as a cluster, both in a positive and negative sense as two sub-factors. This cluster comprises heightened awareness of mental health on the positive side, as well as students' physical health, fatigue, and delays in seeking help on the negative side. Moreover, participants reported trends in student well-being, such as whether it had amplified, stayed stable, decreased, or increased. A university teacher stated at t3 that she was "positively surprised" to see her students coping so well, for example. Whereas participants remained neutral when talking about well-being before the pandemic hit, they got more specific throughout the pandemic, frequently differentiating between negative and positive well-being. Generally though, negative statements about student well-being outweighed the positive ones across all time points, stated mainly by the students.

**Pandemic and society.** Two clusters referred to external circumstances: the pandemic itself and the societal context. For the former, topics like measurement adherence or crisis-related struggles emerged (e.g., quarantine, lockdown, coronavirus anxiety). Regarding the societal context, issues included how the government had handled the crisis and whether society would fall back into old patterns after the pandemic.

**Pandemic-related stressors.** We found all four pandemic-related stressors within the data–particularly at t2 –forming the four sub-factors disruption of social life, uncertainty, daily routines, and financial insecurities. First, the interviewees mentioned disruption of social life as the stressor impacting their well-being most often compared with the remaining three stressors, mainly due to social distancing. International students faced additional social disruption in this regard because of distance and closed borders. Second and third, uncertainty surrounding the pandemic's development and loss of daily routines constituted essential stressors. Regarding the latter, one student emphasised at t4: "it's such a blur, all the things are mashed up together, and I don't know the things I did in months". Still, although participants

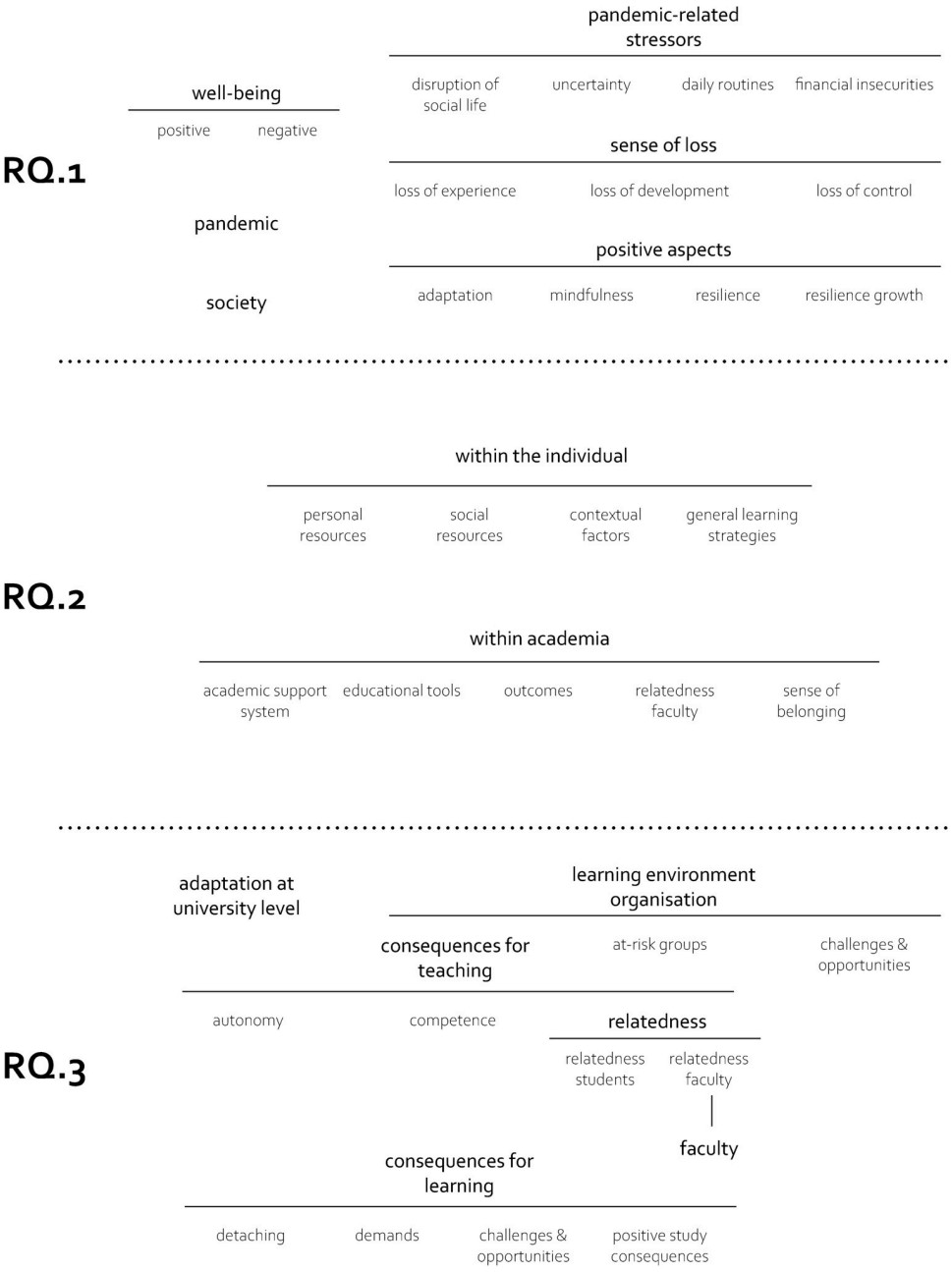

**Fig 2. Code clusters.** This figure includes the codes divided into the three research questions focusing on students' well-being, resilience factors, and learning environment, divided by the dotted lines. The cluster headlines are displayed in boldface.

reported losing structure, they highlighted attempts to regain their daily routines. Lastly, the interviewees mentioned financial insecurities far less frequently than the other stressors.

Considering the longitudinal development of the experiences of these stressors, participants reported disruption of social life, uncertainty, and daily routines throughout the whole pandemic period, yet with particular emphasis shortly after the pandemic hit (t2). At the same time, they reported having adapted to the situation at t3 and t4, respectively, which might explain the decreasing emphasis on the pandemic-related stressors.

**Sense of loss.**   All interviewees described a general sense of loss, which we split into three sub-factors pertaining to the loss of experience, control, and development. Loss of experience was the most prominent across all periods, within and outside the university. One participant noted that life had become "a very very very small [one] of staying in the same place" (student, t2). Regarding the loss of control, the student psychologist noted, "especially for students, . . . there's a lot that they can't control right now" (t2). Therefore, participants reported focusing on what they could control. Just like the pandemic-related stressors, participants highlighted loss of control particularly shortly after the pandemic hit (t2), with less emphasis as it proceeded. Finally, one teacher noted the loss of development: "all my students can pass, get their ECs . . . but [I haven't] challenged them to develop themselves" (t3).

**Positive aspects.**   The interviewees also mentioned positive aspects of the pandemic, including the four sub-factors students' adaptation, mindfulness, resilience, and growth. First, how students adapted to the new situation appeared essential, because "whether they have accepted that or not [had] a big impact" on their well-being (student, t4). As already touched upon, these adaptation processes appeared more evident shortly after the pandemic hit (t2) compared with the other times of measurement. Secondly, the aspect of mindfulness implies slowing down and becoming aware of small things, or as one teacher stated, "the appreciation of what we all have here" (t2). Third, interviewees highlighted reinforced resilience resources due to prior adversity, which led to a strengthened self in the present, and lastly, they commented on their potential to grow their resilience by experiencing the current crisis. This differentiation between resilience due to prior adversity and resilience growth due to the pandemic itself also echoed the longitudinal data: Whereas participants mentioned resilience as a relevant well-being resource already before the pandemic, they only mention the process of resilience growth itself as the pandemic progressed, particularly during the second infection wave (t4). One teacher emphasised:

> Having experienced this is very healthy, actually. . . . Adversity helps you to get to know yourself much better–know yourself under different circumstances, so you get to know your boundaries better, you also get to know . . . about your own flexibility, capability. . . . I also think it humbles you as person . . . and makes you more appreciate- appreciative of what you have. (university teacher, t4)

## RQ.2 resilience factors in times of crisis

The participants' categorisations of resilience factors at t1 led to two main clusters, one within the individual and the other within academia (RQ.2, Fig 2; see S2 File for the raw categorisations). Interestingly, students reported relying more strongly on individual compared with academic resilience factors. Based on their categorisations, participants identified aspects regarding their top three resilience factors (at t1), the decreased or increased importance of those during the pandemic (t2), and their potential for resilience growth (t4; see Table 2) for a longitudinal comparison.

**Within the individual.**   The individual resilience cluster entails four sub-factors: personal and social resources, contextual factors, and general learning strategies. First, students' personal resources included leisure and extracurricular activities (e.g., sports, relaxation) and personal characteristics (e.g., being reflective or self-compassionate, discovering boundaries). Furthermore, participants noted normalising negative emotions as well as understanding that others struggle, too, and that their worth remains independent of their study outcomes.

**Table 2. Classification of resilience factors: Top 3 resilience factors (t1); which ones have become more or less important or changed in nature due to the pandemic (t2); which ones entail the potential for resilience growth.**

| | Student 1 | Student 2 | FM 1 | FM 2 | FM 3 | FM 4 |
|---|---|---|---|---|---|---|
| Top 1 | Personal resources | Social resources | Personal resources; outcome | General learning strategies[a] | Personal resources[a] | Personal resources |
| Top 2 | Personal resources | General learning strategies | Relatedness FM | Personal resources[a] | Personal resources[a] | Social resources |
| Top 3 | Social resources | Personal resources | Relatedness FM | Personal resources[a] | Sense of belonging[a] | Relatedness FM |
| More important | Personal resources | Social resources; general learning strategies; personal resources | Personal resources; relatedness FM | Personal resources; contextual factors | Academic support system; social resources; relatedness FM; personal resources; sense of belonging | Personal resources; social resources |
| Less important | Social resources; academic support system; contextual factors | Academic support system; personal resources | - | Personal resources | Contextual factors; social resources | Academic support system |
| Changed in nature | - | Personal resources | Personal resources; relatedness FM | - | Sense of belonging | Academic support system; social resources; contextual factors; personal resources |
| Potential for resilience growth | Personal resources | Social resources; general learning strategies; personal resources | - | Relatedness FM; personal resources | Personal resources; sense of belonging; academic support system | Personal resources; social resources |

*Note.* [a] Aspects are interchangeable in their ratings; FM = faculty member.

Students' study attitudes, such as not being perfectionistic, also seemed essential as personal resources. Compared with faculty members, students mentioned these personal factors far more frequently. However, they all asserted that personal resources had the greatest potential to result in resilience growth. Secondly, participants emphasised social resources, including both who supported them (e.g., family, friends) and how (e.g., providing study support, socialising opportunities). Throughout all the interviews, students consistently reported social resources when talking about their well-being and resilience. The support staff instead focused more on the third individual resilience factor, contextual factors (e.g., achievement culture, students' surroundings) as essential. Finally, students mentioned a wealth of learning strategies as the fourth individual resilience factor, including prioritising and concentration strategies. Concerning their longitudinal development, personal and social resources seemed of higher relevance in relation to the contextual factors or general learning strategies compared with before the pandemic (t1).

**Within academia.** The academic resilience cluster covers five sub-factors: the academic support system, educational tools, outcomes, relatedness with faculty, and sense of belonging. First, the academic support system constitutes the dominant sub-factor, including faculty, study, and mental health support staff, as identified by the support staff interviewees. In contrast, teachers mentioned educational tools, the second sub-factor, more frequently. The third sub-factor focussed on outcomes, including students' progress, engagement, commitment, and satisfaction. As the fourth sub-factor, the participants mentioned relatedness with faculty, highlighting well-being-promoting interpersonal dynamics. For example, teachers provide students with autonomy, establish open relationships, and help them feel seen as individuals. Whereas university teachers stressed relatedness to faculty members, support staff focused more on students' general sense of belonging, the fifth sub-factor. In a systemic sense, fellow

students, student associations, the faculty, and the university contribute to an academic sense of belonging. Therefore, feeling connected to these institutions could enhance belongingness, whereas disconnection could diminish it. During the pandemic compared with the period shortly before, the academic support system, outcomes, and sense of belonging appeared to lose significance. In contrast, educational tools and relatedness with faculty remained crucial factors also after the pandemic hit.

## RQ.3 the role of the LE in times of crisis

Regarding the LE, we identified a network of clusters surrounding changes within the academic system and the consequences thereof. We identified adaptation at the university level and the learning environment organisation regarding changes within the academic system. Concerning consequences thereof, we differentiate between consequences for teaching–with an emphasis of relatedness and faculty–as well as for learning.

**Adaptation at university level.** The first cluster pertained to the university's adjustment to a remote LE on curricular and organisational levels. Most participants were pleased with how quickly and flexibly the university reacted. However, abrupt cancellations of internships, exams, and lectures led participants to note chaotic adjustments nonetheless.

**Learning environment organisation.** The COVID-19 measures created a mixture of offline, online, and hybrid LE organisation. Notably, students did not believe online teaching worked "as effectively as face-to-face teaching, which is obvious", but they accepted it as the "second-best option" (student, t2). Next to consequences for teaching and learning, we identified two sub-factors within this cluster, namely at-risk groups and challenges and opportunities for the LE. Regarding the latter, interviewees noted that challenges and opportunities surrounding online teaching were closely related and could shape the 'new normal'. Example challenges encompassed mentioned transparent crisis communication and home studies, whereas new opportunities appeared characteristic of the mixture of Les as well. Such challenges could be exacerbated for potential at-risk groups, two of which emerged from our data: first-year and international students. The former likely experienced fewer social opportunities, whereas the latter mainly suffered time differences. Faculty members mentioned at-risk groups far more often than the students, despite one of the students having an international background herself.

**Consequences for teaching.** The cluster regarding the consequences of remote teaching mainly involves the BPN of autonomy, competence, and relatedness. Although external restrictions limited students' autonomy, their ability to schedule time flexibly made "studying and watching the lectures more their own choice and less something you have to do because it's scheduled" (university teacher, t2). Therefore, this flexibility and freedom re-established the initially compromised sense of autonomy and control. In this regard, support staff were aware of both restricted and promoted autonomy as a balancing act. Competence, in contrast, appeared the least affected by the pandemic. The lack of structure and self-discipline seemed to affect students' sense of competence, but experiencing the pandemic also seemed beneficial because they could learn something from the crisis.

Students' sense of relatedness appeared to be the most compromised BPN, emphasising the relevance of social connections during the pandemic. We divided this cluster into relatedness toward fellow students and faculty members. First, students reported their relatedness to fellow students initially in a neutral way; they had made friends and maintained these relationships online. However, both started new studies during the pandemic and highlighted how difficult establishing new relationships could be: "I actually have no idea about other students . . . of my master's, because I don't really talk to them. I don't have any kind of . . . relationship with

them in that sense" (t4). Second, compromised relatedness to the faculty members seemed just as apparent. Specifically at t3 and t4, the growing distance between the two parties was striking. This tendency was also reflected in a decreased ability to take the perspective of the other. Teachers "[didn't] talk enough with students anymore. [They didn't] meet them at the corridors. [They didn't] meet them during break of classes or whatever" (university teacher, t3). Statements such as these highlight faculty members' role in student well-being, as well as their own suffering due to this situation. For instance, teachers noted decreased joy in teaching, though not decreased well-being in general.

**Consequences for learning.** The last cluster pertains to students' learning and differentiates between four sub-factors: detaching, demands, challenges and opportunities for learning, and positive study outcomes. First, students reported detaching from their studies, which led to less engagement and motivation, feelings that increased with time from t2 to t4. They explained, "it [was] difficult . . . to stay motivated, to stay driven". Consequently, they felt "very detached" from their studies, as one student remarked. When asked what she felt connected to, she added:

> I don't really know. Could I answer the course content? [chuckles] I mean, I don't really feel connected to the uni because I've never been there, . . . maybe a few students, I feel a bit connected to because you've had a few seminars together. . . . and maybe, a teacher, but not as a whole. (student, t4)

Second, participants mentioned high study demands. Students realised that the university attempted to maintain the educational quality, though "students [were] really struggling with keeping up the pace" (t2). They suffered from heightened workloads and concentration problems. The retrospective focus group participants associated these two aspects of demands and detaching with relatedness. According to this logic, the impact on one aspect may significantly affect the remaining two as well. Third, students reported challenges and opportunities for learning, mainly involving having to be responsible, flexible, and self-disciplined. Lastly, the interviewees mentioned positive consequences of online education, such as having more time for and pleasure within their studies.

## Discussion

The present study sought to examine students' well-being, resilience factors, and LE during the COVID-19 pandemic. Being longitudinal, the qualitative interview study provides essential details about how students and faculty members perceived the initial months of the pandemic. This long-term, holistic perspective is a strength of this research, and the results add valuable information to the field of academic well-being.

### Student well-being in times of crisis

We found that pandemic-related stressors and the general sense of loss, as noted by Taylor [4] and Holmes et al. [5], influenced students' well-being significantly, particularly by disrupting their social lives. Our results echo prior research stating that social and emotional loneliness strongly predict low pandemic-related well-being and impact students' personal lives [58,59]. However, the loss of daily routines and the uncertainty affected students nearly as much, as noted by prior research as well [59]. Such uncertainties have already been indicated by other studies as well, highlighting the insecurity surrounding returning to on-site teaching and fulfilling their programme successfully [59]. Only financial insecurities were less relevant to the participants. Given that job and, hence, also financial security has been found to predict low

well-being during the COVID-19 pandemic [58], our results may stem from the small and, hence, unrepresentative sample size. In total, all of these stressors have been associated with student well-being throughout the COVID-19 pandemic [60], so that our results add to the wealth of findings already present.

Regarding students' sense of loss, the sub-codes provided interesting insights: the loss of experience was mentioned most frequently, but the loss of development and control also appeared relevant. Such losses echo prior research highlighting the uncertainty of making up for what was lost during these pandemic months and years [59]. Moreover, loss of control emerges as highly relevant for well-being research during COVID-19 as well when combined with uncertainty, also regarding coping and resilience [25,26,61]. Pandemic-related aspects such as uncertainty and loss of control resonate with aspects of traumatic experiences. Previous research similarly locates the COVID-19 pandemic within the context of traumatic psychological responses within and outside academia [25,27]. Identifying the COVID-19 pandemic as a potential collective traumatic experience highlights the role of resilience as a psychological counter-process that should be included in pandemic-related well-being research.

## Resilience factors in times of crisis

Based on our findings, individual and academic resilience factors emerged. Individual resilience factors included personal and social resources, contextual factors, and general learning strategies; those on an academic level comprised the academic support system, educational tools, outcomes, relatedness with faculty members, and sense of belonging. Up until now, researchers have distinguished resilience factors in different ways; for instance, Ang et al. [23] differentiates internal–such as personal resources - and external factors–such as social resources. Yet, taken together, all these resilience factors appeared essential to cope with pandemic-related challenges, as proposed by prior literature as well [23]. All participants rated personal resources consistently as very important. Yet they varied in their perceptions of which other resilience factors were most important: students mainly mentioned social resources concerning relationships outside the academic realm, but university staff considered relatedness with faculty members within academia as more relevant. At this point, we want to acknowledge that both codes overlap, as social resources outside academia may be fellow students that have become friends students meet up with outside university as well. However, we still consider it important to distinguish between the two aspects, as they developed differently throughout the pandemic. For instance, faculty members believed social resources outside academia to be increasingly relevant, whilst relatedness with faculty members lost significance. This perception resonates with previous research noting the crucial relevance of social resources, for both traumatic event recovery and non-pathological academic well-being [23,62]. Likewise, the student-faculty relationship has been found to promote student well-being already before the pandemic [63] and deserves attention despite the consistent impairments throughout the pandemic. As more resilient students reported better attitudes towards online and hybrid education, investing in these resilience factors appears essential, particularly in times of remote teaching [15].

Beyond that, the participants rated these factors regarding their potential for post-traumatic or resilience growth. They indicated particularly personal but also social resources have grown because of the pandemic. COVID-19-related research has already started investigating the effects surrounding post-traumatic growth in the aftermath of the pandemic [61]. For instance, Vazquez and colleagues [64] found post-traumatic growth associated with beliefs about a good world, openness to the future, and identification with humanity. At the same time, not being

able to cope with uncertainty, a pandemic-related stressor that we found in our results as well, led to post-traumatic stress instead. Moreover, Baños and colleagues [65] reported positive functioning decreased throughout the pandemic, whereas at the same time, emotional distress, but also personal strength increased. Taken together, such findings hint towards the great potential for post-traumatic growth due to having experienced the COVID-19 pandemic, particularly in terms of personal and social resources.

## The role of the LE in times of crisis

Lastly, we focused on the link between the LE and students' need satisfaction. Of these, competence appeared to have been affected the least. For autonomy, the pattern that emerged distinguishes experiences within and outside academia. Outside academia, external restrictions imposed by pandemic-related measures impaired students' feelings of autonomy. Within academia, their autonomy increased due to flexibility and independence through remote educational settings. The focus group echoed this duality when a teacher expressed the difficulties of finding a good balance between giving students more autonomy or more structure when teaching online.

Relatedness emerged as especially crucial for students' well-being in pandemic times. Given the social restriction measures, a compromised sense of relatedness and connection should be expected; the findings pertaining to growing distances confirm this hypothesis and indicate that relatedness has become increasingly disrupted. Students felt disconnected and unrelated to their remote educational system, which may lead to lower adaptability towards changes within their LE [43]. Disconnection among students and with teachers in online teaching settings has been addressed by prior research, both during and before the pandemic [51,59]. On the one hand, student-teacher relationships appear to have become deeper through empathy and flexibility to ease the students' situation [59]. On the other hand, studies have shown that students in online teaching programs tend to disengage from peers and actively withdraw from online social activities [66]. Such behaviour may amplify social disconnections within the academic system even further. Our data indicated that participants connected this lack of relatedness with feelings of detachedness and academic demands. These findings echo other studies highlighting the relationship between lacking relatedness and unfavourable learning outcomes during the COVID-19 pandemic [67]. The participants in the focus group, furthermore, noted differences in relatedness depending on whether students started before or during online teaching: maintaining a sense of connectedness in an online environment appears less of a concern than establishing it in the first place. Prior research has investigated this phenomenon as well, distinguishing between deeper connections with good friends and increased disconnection from fellow students [59].

Ultimately, we found both negative and positive consequences for online learning, revealing the complex aftermath of the pandemic that academia must take into account. Despite positive aspects, previous research has demonstrated that students generally prefer offline to online education [54] and that studying online is unfavourable for their resilience development [68]. Other researchers investigating academic well-being throughout COVID-19 have found this duality as well [59,69]. According to these findings, advantages encompass shorter travel time, easier access to education and the possibility to study according to one's personal learning habits. At the same time, disadvantages span depersonalisation, screen fatigue, and technical difficulties [59,69,70]. Therefore, universities must attempt to balance the positive and negative consequences that online teaching entails.

## Limitations

Although the findings contribute to a deeper understanding of the interaction between students' well-being and the LE, certain limitations remain. These mainly regard the rather small sample. Interviewing only two students may have excluded essential aspects, such as a precarious financial situation. Also in general, a sample size of only six participants has most likely restricted the representability of this study, as it encompasses too few individual experiences. A sample size of 15 to 20 interviewees would be preferable for qualitative research, though prior research also indicates that 12 participants can suffice and achieve saturation when using thematic coding, and as few as 6 participants may already cover essential elements [71]. For our study, the six interviewees participated four times throughout the study, adding up to 24 interviews in total. Consequently, the present findings may be grounded in sufficient data to ensure basic saturation regardless. Moreover, the qualitative nature of the study in itself may have been an additional limitation. Whereas the level of detail of individual experiences belongs to the strengths of qualitative research, it automatically comes with a lack of generalisability to a greater population.

Another limitation involves the interviewees' willingness to participate. The interviewees expressed interest in the topic prior to the study and enough enthusiasm to participate for nearly 12 months. This recruitment method creates a potential bias through, by focusing on people who are specifically aware of and prone to engage with well-being; we might have excluded the perspectives of students and faculty members unaware of these topics. Finally, interviewees reported that participating in the interview study had promoted their well-being. Therefore, the research itself may have affected them, and the results may be more favourable than they would have been otherwise.

## Implications and further research

The present findings support the importance of the BPN for academic well-being research, highlighting the link between the self-determination theory and the academic system. We recommend that research on this topic continues, both qualitatively and quantitatively. Furthermore, the current study raises a variety of potential implications for how the LE could enhance student well-being during distance learning. Particularly during the focus group, the participants reflected on what our preliminary results could mean for an educational "new normal". They emphasised best practices that they had already experienced during remote teaching, such as using annotations combined with asynchronous recordings, so teachers and students become pen-pals. Furthermore, engaging in written instead of verbal conversation led to more openness involving sensitive issues so that quiet students participated more. Therefore, such educational practices may be beneficial.

Satisfying students' BPN appeared particularly relevant for educational practice. Investing in need satisfaction is crucial for developing students' resilience in online LEs [72,73], as evidenced by interventions to promote students' BPN in prior research [74]. Moreover, the participants suggested additional educational tools to satisfy various BPN simultaneously. One example constitutes small group work, as emphasised by other researchers as well [59]. Letting students choose their groups themselves may stimulate their feeling of autonomy and relatedness, as such a process gives them the feeling of control while allowing a safe shared space within these groups.

Participants also suggested ideas to promote students' sense of autonomy specifically. The LE could be designed to counteract students' uncertainty and loss of control. For instance, being transparent in academic decisions and curricula changes may prevent students from feeling helpless or uncertain. Furthermore, participants proposed more flexibility for

deadlines, online exams, and the freedom to choose an examination mode themselves, at least for more experienced students. Giving students a participatory voice in shaping their LE may enhance their feeling of control [46]. However, giving them flexibility may be counterproductive if the external structure is lacking. Therefore, teachers should juggle providing structure while simultaneously emphasising students' autonomy.

Investing in both student–teacher and student-student relationships may be valuable for students' sense of relatedness. Research has already called for a reconceptualization of the relatedness concept for higher education in post-pandemic times [75]. Providing time to connect might constitute a first step towards a more connected academic system. Actively investing in relationships is critical though, when conventional means to establish and maintain relationships (e.g., meeting in the hallway) are lacking. Instead, teachers might ask students how they are doing in one-to-one or group settings. Furthermore, new opportunities to connect could arise from small-scale teaching and steady study or mentor groups. Staying together in one fixed study group throughout the class trajectory could contribute to a sense of relatedness. In addition, students reported that they were more engaged and felt more connected to others when seeing them on video. Considering that turning one's video off has become somewhat standard, reversing this practice would be a quick and easy way to improve students' relatedness. Teachers sharing informal information about themselves also may result in the sense of cohesion and encourage students to open up, too. Beyond that, student and study organisations could serve as intermediaries to re-engage students with their study surroundings.

The feeling of decreased relatedness between teachers and students is particularly concerning. This phenomenon can arise from non-transparent communication on both sides: Students may not be aware of the teachers' efforts in online teaching, and teachers may have lost sense of their students' well-being. Teachers experienced remote education in the pandemic negatively as well [59,76,77], and communicating such struggles to students may create a sense of common humanity. The feeling of being in it together and acknowledging one another's problems could then contribute to a greater feeling of relatedness. Therefore, openness and appreciation on both sides could be a first step toward a more related academic environment.

## Conclusion

The current study aimed to investigate university students' well-being, resilience factors, and learning environment. Whereas the pandemic resulted in students becoming more mindful, it mainly influenced their well-being negatively by disrupting their social life and daily routines. However, students possessed a wealth of resilience factors that helped them face such adversities–particularly personal resources on an individual and the academic support system on an institutional level. Lastly, the present findings emphasise how the learning environment contributes to satisfying students' basic psychological needs and, therefore, how it impacts their well-being throughout pandemic times. Particularly their needs for autonomy and relatedness seemed compromised. Therefore, investing in satisfying these needs through the participants' various suggestions could facilitate the process of reshaping the academic system to become a healthy and thriving 'new normal'.

## Supporting information

**S1 File. Supporting information I–materials.** Including the interview guides and the diary template, (Table S1-S6).
(PDF)

**S2 File. Supporting information II–results.** Including the categorisations identified during the interviews and discussed during the focus group (Table Figure S1 & S2). (PDF)

**S3 File. Supporting information III–codebook.** Including the final codebook (Table S7). (PDF)

## Acknowledgments

We thank Friederike Axmann for her help in analysing the data as an independent coder.

## Author Contributions

**Conceptualization:** Lisa Kiltz, Marjon Fokkens-Bruinsma, Ellen P. W. A. Jansen.

**Data curation:** Lisa Kiltz.

**Formal analysis:** Lisa Kiltz, Marjon Fokkens-Bruinsma, Ellen P. W. A. Jansen.

**Investigation:** Lisa Kiltz.

**Methodology:** Lisa Kiltz, Marjon Fokkens-Bruinsma, Ellen P. W. A. Jansen.

**Project administration:** Lisa Kiltz, Marjon Fokkens-Bruinsma, Ellen P. W. A. Jansen.

**Supervision:** Marjon Fokkens-Bruinsma, Ellen P. W. A. Jansen.

**Writing – original draft:** Lisa Kiltz.

**Writing – review & editing:** Lisa Kiltz, Marjon Fokkens-Bruinsma, Ellen P. W. A. Jansen.

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
