## [Decision Letter · Decision Letter 0]

12 Dec 2022

TELÉFONO-D-22-17942Atrapados entre el alivio y la inquietud: cómo se relaciona el bienestar de los estudiantes universitarios con su entorno de aprendizaje durante la pandemia de COVID-19 en los Países BajosPLOS ONE 

Estimado Dr. Lisa Kiltz,

Gracias por enviar su manuscrito a PLOS ONE. Después de una cuidadosa consideración, creemos que tiene mérito pero no cumple completamente con los criterios de publicación de PLOS ONE en su forma actual. Por lo tanto, lo invitamos a enviar una versión revisada del manuscrito que aborde los puntos planteados durante el proceso de revisión. 

We look forward to receiving your revised manuscript.

Kind regards,

Teresa Pozo-Rico

Academic Editor

PLOS ONE

Journal Requirements:

2. Peer review at PLOS ONE is not double-blinded (https://journals.plos.org/plosone/s/editorial-and-peer-review-process). For this reason, authors should include in the revised manuscript all the information removed for blind review.

* Please provide additional details regarding ethical approval in the body of your manuscript. In the Methods section, please ensure that you have specified the name of the IRB/ethics committee that approved your study.

Reviewers' comments:

Reviewer's Responses to Questions

**Comments to the Author**

1. Is the manuscript technically sound, and do the data support the conclusions?

Reviewer #1: Yes

Reviewer #2: Partly

2. Has the statistical analysis been performed appropriately and rigorously? 

Reviewer #1: Yes

Reviewer #2: N/A

3. Have the authors made all data underlying the findings in their manuscript fully available?

Reviewer #1: Yes

Reviewer #2: Yes

4. Is the manuscript presented in an intelligible fashion and written in standard English?

Reviewer #1: Yes

Reviewer #2: Yes

5. Review Comments to the Author

Reviewer #1: Thank you for giving me the chance to review the manuscript "Caught between relief and unease: How university students’ well-being relates to their learning environment during the COVID-19 pandemic in the Netherlands:. The authors did a great job in their analysis and writing. I have very minor comments:

Introduction:

I don’t necessarily agree with line 79. There have been ample studies published on resilience during COVID-19, that target different populations or groups of individuals. To mention few:

Barzilay, R., Moore, T.M., Greenberg, D.M. et al. Resilience, COVID-19-related stress, anxiety and depression during the pandemic in a large population enriched for healthcare providers. Transl Psychiatry 10, 291 (2020). https://doi.org/10.1038/s41398-020-00982-4

Labrague, LJ, de los Santos, JA. COVID-19 anxiety among front-line nurses: Predictive role of organisational support, personal resilience and social support. J Nurs Manag. 2020; 28: 1653– 1661. https://doi.org/10.1111/jonm.13121

Ye, Z., Yang, X., Zeng, C., Wang, Y., Shen, Z., Li, X. and Lin, D. (2020), Resilience, Social Support, and Coping as Mediators between COVID-19-related Stressful Experiences and Acute Stress Disorder among College Students in China. Appl Psychol Health Well-Being, 12: 1074-1094. https://doi.org/10.1111/aphw.12211

Haldane, V., De Foo, C., Abdalla, S.M. et al. Health systems resilience in managing the COVID-19 pandemic: lessons from 28 countries. Nat Med 27, 964–980 (2021). https://doi.org/10.1038/s41591-021-01381-y

Prime, H., Wade, M., & Browne, D. T. (2020). Risk and resilience in family well-being during the COVID-19 pandemic. American Psychologist, 75(5), 631–643. https://doi.org/10.1037/amp0000660

Discussion:

The discussion would benefit from mentioning the findings of:

Ang, W., Shorey, S., Lopez, V. et al. Generation Z undergraduate students’ resilience during the COVID-19 pandemic: a qualitative study. Curr Psychol (2021). https://doi.org/10.1007/s12144-021-01830-4

Prieto, D., Tricio, J., Cáceres, F., Param, F., Meléndez, C., Vásquez, P. and Prada, P. (2021), Academics' and students' experiences in a chilean dental school during the COVID-19 pandemic: A qualitative study. Eur J Dent Educ, 25: 689-697. https://doi.org/10.1111/eje.12647

Other comments:

Line 69: may “Counteract” - please ensure all text is checked for spelling and grammar

Figure 1: what does UG stand for?

Reviewer #2: In this qualitative study the authors attempted to assess whether the COVID-19 pandemic affected on students’ well-being, relevant resilience factors and the role of the learning environment in this regard.

They enrolled six participants who were administered a semi-structured guide at four different time, founding evidence of the impact of the pandemic on all the three aspects investigated. Hence, the authors suggest a reshape of the academic system according to their findings.

The article is interesting and may provide useful information, but I suggest some revisions:

1-I suggest revising the methodology section. It would be useful to specify more in details the time period in which participants were recruited and the timing with which the semi-structured interview was administered (not only in Figure 1). Likewise, it would be important to make a summary of the tables found in the supporting information in the methodology paragraph, to better understand what it was asked to the participants in the different timeframes.

2-In my opinion, it is necessary to explain in more detail the limitations of the study. First, it is important to point out what are the weaknesses of a qualitative study. Secondly, I suggest that more emphasis should be placed on the smallness of the sample and all that goes with it.

3-Lines 444-444: “Therefore, universities must attempt to balance the positive and negative consequences that online eaching entails”. It would be helpful to explain in further detail the positive and negative aspects of online education with a more comprehensive survey of the available literature.

6. PLOS authors have the option to publish the peer review history of their article (what does this mean?). If published, this will include your full peer review and any attached files.

Reviewer #1: No

Reviewer #2: No

While revising your submission, please upload your figure files to the Preflight Analysis and Conversion Engine (PACE) digital diagnostic tool, https://pacev2.apexcovantage.com/. PACE ayuda a garantizar que las cifras cumplan con los requisitos de PLOS. Para utilizar PACE, primero debe registrarse como usuario. El registro es gratuito. Luego, inicie sesión y navegue a la pestaña CARGAR, donde encontrará instrucciones detalladas sobre cómo usar la herramienta. Si encuentra algún problema o tiene alguna pregunta al usar PACE, envíe un correo electrónico a PLOS a  figures@plos.org . Tenga en cuenta que los archivos de información de respaldo no necesitan este paso.

---

## [Author Response · Author response to Decision Letter 0]

23 Jan 2023

Reviewer #1: 

Thank you for giving me the chance to review the manuscript "Caught between relief and unease: How university students’ well-being relates to their learning environment during the COVID-19 pandemic in the Netherlands:. The authors did a great job in their analysis and writing. I have very minor comments:

Introduction:

I don’t necessarily agree with line 79. There have been ample studies published on resilience during COVID-19, that target different populations or groups of individuals. To mention few:

Barzilay, R., Moore, T.M., Greenberg, D.M. et al. Resilience, COVID-19-related stress, anxiety and depression during the pandemic in a large population enriched for healthcare providers. Transl Psychiatry 10, 291 (2020). https://doi.org/10.1038/s41398-020-00982-4

Labrague, LJ, de los Santos, JA. COVID-19 anxiety among front-line nurses: Predictive role of organisational support, personal resilience and social support. J Nurs Manag. 2020; 28: 1653– 1661. https://doi.org/10.1111/jonm.13121

Ye, Z., Yang, X., Zeng, C., Wang, Y., Shen, Z., Li, X. and Lin, D. (2020), Resilience, Social Support, and Coping as Mediators between COVID-19-related Stressful Experiences and Acute Stress Disorder among College Students in China. Appl Psychol Health Well-Being, 12: 1074-1094. https://doi.org/10.1111/aphw.12211

Haldane, V., De Foo, C., Abdalla, S.M. et al. Health systems resilience in managing the COVID-19 pandemic: lessons from 28 countries. Nat Med 27, 964–980 (2021). https://doi.org/10.1038/s41591-021-01381-y

Prime, H., Wade, M., & Browne, D. T. (2020). Risk and resilience in family well-being during the COVID-19 pandemic. American Psychologist, 75(5), 631–643. https://doi.org/10.1037/amp0000660

Dear reviewer, first of all thank you for taking the time to review our manuscript and being positive about it! 

Thanks for pointing us to more literature about resilience during COVID-19. We agree that the sentence no longer holds and deleted it instead. Additionally, we decided to add the study by Barzilay et al. (2020) as we considered it suitable for our study’s context; the study by Ye et al. (2020) was indeed already cited in the paragraph. 

For instance, more resilient students perceive less stress during COVID-19.[7] Likewise, students’ resilience moderates pandemic-related stressful experiences and acute stress disorder.[26] Also in the general population, more resilient individuals experienced fewer worries, anxiety, or depression symptoms. [27]

Discussion:

The discussion would benefit from mentioning the findings of:

Ang, W., Shorey, S., Lopez, V. et al. Generation Z undergraduate students’ resilience during the COVID-19 pandemic: a qualitative study. Curr Psychol (2021). https://doi.org/10.1007/s12144-021-01830-4

Prieto, D., Tricio, J., Cáceres, F., Param, F., Meléndez, C., Vásquez, P. and Prada, P. (2021), Academics' and students' experiences in a chilean dental school during the COVID-19 pandemic: A qualitative study. Eur J Dent Educ, 25: 689-697. https://doi.org/10.1111/eje.12647

Thank you for your nice suggestion. We read the two papers and agree that they are very fitting and relevant for our study and incorporated them in our discussion (see revised manuscript).

Other comments:

Line 69: may “Counteract” - please ensure all text is checked for spelling and grammar 

Thank you for making us aware of this mistake. We corrected it and additionally checked the whole document once again using a spelling and grammar tool and hope that it is now without any language errors. 

Figure 1: what does UG stand for?

UG stands for the University of Groningen, which we anonymized for the first round of revisions. We now added a respective explanation to the note of the figure, so that it should be clearer now. 

Again thank you very much for your help regarding our manuscript!

 

Reviewer #2: 

In this qualitative study the authors attempted to assess whether the COVID-19 pandemic affected on students’ well-being, relevant resilience factors and the role of the learning environment in this regard.

They enrolled six participants who were administered a semi-structured guide at four different time, founding evidence of the impact of the pandemic on all the three aspects investigated. Hence, the authors suggest a reshape of the academic system according to their findings.

The article is interesting and may provide useful information, but I suggest some revisions:

1-I suggest revising the methodology section. It would be useful to specify more in details the time period in which participants were recruited and the timing with which the semi-structured interview was administered (not only in Figure 1). Likewise, it would be important to make a summary of the tables found in the supporting information in the methodology paragraph, to better understand what it was asked to the participants in the different timeframes. 

Dear reviewer; first of all thank you for your generally positive feedback and the time you took to give suggestions to improve the manuscript. 

We agree that more specific information about the time period would be useful to understand the context the study is set in, which is why we included Figure 1 in the first place. We have now elaborated on this and also added some information in written form: 

As the situation changed quickly during these months, it appears essential to understand the situational context (see Figure 1). We recruited the participants at the end of 2019 and interviewed them for the first time before the pandemic hit the Netherlands. The second time of data collection then fell in the period of the first lockdown and the end of the academic year, with social distancing measures and the closure of universities in place. The third interviews, however, took place in the summer holidays, during which normality had somewhat returned, although remaining socially restricted. Lastly, t4 was similarly affected as t2 by the second lockdown in the Netherlands whilst the new academic year had just begun.

Beyond that, we agree that additional information concerning the interview schedules improves the manuscript and added a respective summarised table to the main text as well. 

2-In my opinion, it is necessary to explain in more detail the limitations of the study. First, it is important to point out what are the weaknesses of a qualitative study. Secondly, I suggest that more emphasis should be placed on the smallness of the sample and all that goes with it. 

Thank you for the remark. We agree that more emphasis on the limitations helps understand the interpretability of the study better. This is why we expanded this section:

Although the findings contribute to a deeper understanding of the interaction between students’ well-being and the LE, certain limitations remain. These mainly regard the rather small sample. Interviewing only two students may have excluded essential aspects, such as a precarious financial situation. Also in general, a sample size of only six participants has most likely restricted the representability of this study, as it encompasses too few individual experiences. A sample size of 15 to 20 interviewees would be preferable for qualitative research, though prior research also indicates that 12 participants can suffice and achieve saturation when using thematic coding, and as few as 6 participants may already cover essential elements.[65] For our study, the six interviewees participated four times throughout the study, adding up to 24 interviews in total. Consequently, the present findings may be grounded in sufficient data to ensure basic saturation regardless. Moreover, the qualitative nature of the study in itself may have been an additional limitation. Whereas the detailedness of individual experiences belongs to the strengths of qualitative research, it automatically comes with a lack of generalisability to a greater population.

3-Lines 444-444: “Therefore, universities must attempt to balance the positive and negative consequences that online eaching entails”. It would be helpful to explain in further detail the positive and negative aspects of online education with a more comprehensive survey of the available literature. 

Thank you for the remark and we agree that the paragraph benefits from more detail. This is why we added additional literature to the paragraph to elaborate on both the positive and negative consequences of online teaching. 

Ultimately, we found both negative and positive consequences for online learning, revealing the complex aftermath of the pandemic that academia must take into account. Despite positive aspects, previous research has demonstrated that students generally prefer offline to online education,[52] and that studying online is unfavourable for their resilience development.[64] Other researchers investigating academic well-being throughout COVID-19 have found this duality as well.[Ferrer,Prieto] According to these findings, advantages encompassed shorter travel time, easier access to education and increased student involvementthe possibility to study according to one’s personal learning style., whereasAt the same time, disadvantages spanned depersonalisation, screen fatigue, and technical difficulties.[Eringfeld,Ferrer,Prieto] Therefore, universities must attempt to balance the positive and negative consequences that online teaching entails.

Thanks again for your time and effort to help us improve this manuscript!

---

## [Decision Letter · Decision Letter 1]

19 Jun 2023

PONE-D-22-17942R1Caught between relief and unease: How university students’ well-being relates to their learning environment during the COVID-19 pandemic in the NetherlandsPLOS ONE

Dear Dr. Kiltz,

Thank you for submitting your manuscript to PLOS ONE. After careful consideration, we feel that it has merit but does not fully meet PLOS ONE’s publication criteria as it currently stands. Therefore, we invite you to submit a revised version of the manuscript that addresses the points raised during the review process.

We look forward to receiving your revised manuscript.

Kind regards,

Fatma Refaat Ahmed, Ph.D.

Academic Editor

PLOS ONE

Reviewers' comments:

Reviewer's Responses to Questions

**Comments to the Author**

1. If the authors have adequately addressed your comments raised in a previous round of review and you feel that this manuscript is now acceptable for publication, you may indicate that here to bypass the “Comments to the Author” section, enter your conflict of interest statement in the “Confidential to Editor” section, and submit your "Accept" recommendation.

Reviewer #3: (No Response)

2. Is the manuscript technically sound, and do the data support the conclusions?

Reviewer #3: Partly

3. Has the statistical analysis been performed appropriately and rigorously? 

Reviewer #3: N/A

4. Have the authors made all data underlying the findings in their manuscript fully available?

Reviewer #3: Yes

5. Is the manuscript presented in an intelligible fashion and written in standard English?

Reviewer #3: Yes

6. Review Comments to the Author

Reviewer #3: The manuscript "Caught between relief and unease: How university students’ well-being relates to their learning environment during the COVID-19 pandemic in the Netherlands” has an interesting topic and can bring new insights about factors contributing to university students’ well-being and the development of students’ resilience during the pandemic. While the overall writing is excellent, there are some issues that need to be addressed.

First, because of the very small sample size the study should more strongly focus on the longitudinal data. Especially a comparison of pre-COVID-data with data collected during the pandemic might be very informative.

Second, the presentation of the results could be optimized:

Table 2: I was irritated because the text uses a different wording from the table. Are personal resources (as mentioned in the text) and individual resources equivalent?

In this same vein: I am missing a clear description of the outcome-categories/sub-factors as well as a delimitation of the categories from each other. There seems to be some overlap between the categories/sub-factors. This makes the results hard to understand. There is a somewhat clearer description in the discussion section, but an exact description of the categories/sub-factors needs to be provided in the results-section.

I am also missing information concerning the development of resilience that go beyond the participants explicit statements but may be evident from the longitudinal data.

Minor issues:

p.1, 48-49: “For instance, governments introduced persisting restrictions on social and public life that have substantially reshaped the academic learning environment (LE), defined as stakeholders, and educational and structural aspects” Is “[…] defined as stakeholders, and educational and structural aspects” the definition of an academic learning environment? If so, this is not readily intelligible. Perhaps you can find a wording that is more easily understandable.

p.2, 77-78 I believe this sentence should be in the past tense, as is the following sentence.

p.6 128- 130 “Many students thus have experienced decreased motivation and engagement (lack of autonomy), unproductiveness, mental overload (lack of competence), as well as isolation (lack of relatedness).” The relation of the bracket expressions to the text is not obvious. I think that the authors mean that decreases in motivation is due to a lack of autonomy support and so on. Please specify, to make this obvious for the reader.

p.6 130-131 “Students’ relationships with peers and teachers have especially suffered.[53,54]” You might add a quantitative study here, e.g.

Müller, F.H., Thomas, A.E., Carmignola, M., Dittrich, A.-K., Eckes, A., Großmann, N., Martinek, D., Wilde, M. & Bieg, S. (2021) University Students’ Basic Psychological Needs, Motivation, and Vitality Before and During COVID-19: A Self-Determination Theory Approach. Frontiers in Psychology, 12:775804.

p.20 410-411 “students mainly mentioned social resources, but university staff considered relatedness with faculty members as more relevant.“ I wonder why relatedness with faculty members is not considered a social resource but considered an academic resilience factor.

p.21 439-441 “Although a compromised sense of relatedness and connection should be expected, given social restriction measures, the findings pertaining to growing distances indicate that relatedness has become increasingly disrupted.” This sentence is illogical – starting with ‘although’ I would expect a different outcome instead of findings that corroborate the expectations.

p.22 475 “For our study, the six interviewees participated four times throughout the study, adding up to 24 interviews in total.” contradicts p.11 224-226 “To ensure interrater reliability, two independent researchers executed this coding stage, one for the whole data set and the other for 8 of the 20 interviews.”

p.23 “As most studies in this vein have been mainly qualitative,[46] researchers should pursue more quantitative insights, to generalize the results.” I cannot agree with this statement as there are numerous quantitative studies on this topic. You can find many of these papers on https://selfdeterminationtheory.org/topics/application-covid19/

7. PLOS authors have the option to publish the peer review history of their article (what does this mean?). If published, this will include your full peer review and any attached files.

Reviewer #3: No

---

## [Author Response · Author response to Decision Letter 1]

24 Aug 2023

Reviewer #3: 

The manuscript "Caught between relief and unease: How university students’ well-being relates to their learning environment during the COVID-19 pandemic in the Netherlands” has an interesting topic and can bring new insights about factors contributing to university students’ well-being and the development of students’ resilience during the pandemic. While the overall writing is excellent, there are some issues that need to be addressed.

First, because of the very small sample size the study should more strongly focus on the longitudinal data. Especially a comparison of pre-COVID-data with data collected during the pandemic might be very informative.

- Dear reviewer, first of all thank you for taking the time to review our manuscript and being positive about it! 

We agree that the longitudinal data constitutes a strong point of our study and emphasized the findings surrounding the developments throughout the pandemic. Therefore, we added additional longitudinal information that we omitted previously due to the word count. For instance: 

"Moreover, participants reported trends in student well-being, such as whether it had amplified, stayed stable, decreased, or increased. A university teacher stated at t3 that she was “positively surprised” to see her students coping so well, for example. Whereas participants remained neutral when talking about well-being before the pandemic hit, they got more specific throughout the pandemic, frequently differentiating between negative and positive well-being. Generally though, negative statements about student well-being outweighed the positive ones across all time points, stated mainly by the students." 

and:

"Considering the longitudinal development of the experiences of these stressors, participants reported disruption of social life, uncertainty, and daily routines throughout the whole pandemic period, yet with particular emphasis shortly after the pandemic hit (t2). At the same time, they reported having become used to and having accepted the situation at t3 and t4, respectively, which might explain the decreasing emphasis on the pandemic-related stressors." 

Second, the presentation of the results could be optimized:

Table 2: I was irritated because the text uses a different wording from the table. Are personal resources (as mentioned in the text) and individual resources equivalent?

- We apologise for the ambiguities. The table should indeed display ‘personal resources’ instead of ‘individual resources’ – we adapted the table accordingly. 

In this same vein: I am missing a clear description of the outcome-categories/sub-factors as well as a delimitation of the categories from each other. There seems to be some overlap between the categories/sub-factors. This makes the results hard to understand. There is a somewhat clearer description in the discussion section, but an exact description of the categories/sub-factors needs to be provided in the results-section.

- We agree that the results are complicated to disentangle, particularly due to the overlap between categories noted by the reviewer. We now rewrote the section for more clarity and added sub headings to structure it better. For instance: 

"For student well-being, we identified well-being itself as a cluster, both in a positive and negative sense as two subfactors. This cluster comprises heightened awareness of mental health on the positive side, as well as students’ physical health, fatigue, and delays in seeking help on the negative side."

Additionally, we added a clarification paragraph at the beginning of the results section: 

"In the following, we will give an overview over the clusters we identified throughout the analysis, structured according to the three research questions. We differentiate between clusters – an overall theme of various codes – and subfactors – more detailed themes we found within these clusters. Figure 2 depicts an overview of these clusters (displayed in bold) and potential subfactors (displayed in regular font)."

I am also missing information concerning the development of resilience that go beyond the participants explicit statements but may be evident from the longitudinal data.

- We tried to incorporate a more elaborate comment on the longitudinal development of resilience beyond the participants’ statement: 

"Third, interviewees highlighted reinforced resilience resources due to prior adversity, which led to a strengthened self in the present, and lastly, they commented on their potential to grow their resilience by experiencing the current crisis. This differentiation between resilience due to prior adversity and resilience growth due to the pandemic itself also echoed the longitudinal data: Whereas participants mentioned resilience as a relevant well-being resource already before the pandemic, they only mention the process of resilience growth itself as the pandemic progressed, particularly during the second infection wave (t4)."

We hope that our adjustments in general and specifically for the development of resilience clarify and emphasise the longitudinal results better!

Minor issues:

p.1, 48-49: “For instance, governments introduced persisting restrictions on social and public life that have substantially reshaped the academic learning environment (LE), defined as stakeholders, and educational and structural aspects” Is “[…] defined as stakeholders, and educational and structural aspects” the definition of an academic learning environment? If so, this is not readily intelligible. Perhaps you can find a wording that is more easily understandable.

- We adjusted the passage and hope that it is now clearer what exactly we meant by it: 

"For instance, governments introduced persisting restrictions on social and public life that have substantially reshaped the academic learning environment (LE). For the scope of the current study, we define the LE as an environment in which stakeholders act (e.g., students, teachers, and support staff), and in which educational (e.g., teaching modes) and structural aspects (e.g., support systems) play crucial roles. As university students, hence, faced unexpected remote teaching, social distancing, and lockdown within their LE, their well-being and resilience resources may have been impacted."

p.2, 77-78 I believe this sentence should be in the past tense, as is the following sentence.

- That is true, thank you for pointing it out. We adjusted the sentences accordingly.

p.6 128- 130 “Many students thus have experienced decreased motivation and engagement (lack of autonomy), unproductiveness, mental overload (lack of competence), as well as isolation (lack of relatedness).” The relation of the bracket expressions to the text is not obvious. I think that the authors mean that decreases in motivation is due to a lack of autonomy support and so on. Please specify, to make this obvious for the reader.

- We adjusted the passage to clarify that these aspects lead to a lack of need satisfaction: 

"These aspects may add up to a lack of need satisfaction: Many students have experienced decreased motivation and engagement (corresponding to a lack of autonomy), unproductiveness, mental overload (corresponding to a lack of competence), as well as isolation (corresponding to a lack of relatedness).[18,52]"

p.6 130-131 “Students’ relationships with peers and teachers have especially suffered.[53,54]” You might add a quantitative study here, e.g.

Müller, F.H., Thomas, A.E., Carmignola, M., Dittrich, A.-K., Eckes, A., Großmann, N., Martinek, D., Wilde, M. & Bieg, S. (2021) University Students’ Basic Psychological Needs, Motivation, and Vitality Before and During COVID-19: A Self-Determination Theory Approach. Frontiers in Psychology, 12:775804.

- Thank you for pointing out the study by Müller et al; it is very fitting indeed! We added it to the manuscript and the reference list. 

p.20 410-411 “students mainly mentioned social resources, but university staff considered relatedness with faculty members as more relevant.“ I wonder why relatedness with faculty members is not considered a social resource but considered an academic resilience factor.

- We understand the confusion and tried to clarify the passage to make the differentiation between the two codes clearer, while acknowledging that there is some overlap between them: 

"Yet they varied in their perceptions of which other resilience factors were most important: students mainly mentioned social resources concerning relationships outside the academic realm, but university staff considered relatedness with faculty members within academia as more relevant. At this point, we want to acknowledge that both codes overlap, as social resources outside academia may be fellow students that have become friends students meet up with outside university as well. However, we still consider it important to distinguish between the two aspects, as they developed differently throughout the pandemic. For instance, faculty members believed social resources outside academia to be increasingly relevant, whilst relatedness with faculty members lost significance."

p.21 439-441 “Although a compromised sense of relatedness and connection should be expected, given social restriction measures, the findings pertaining to growing distances indicate that relatedness has become increasingly disrupted.” This sentence is illogical – starting with ‘although’ I would expect a different outcome instead of findings that corroborate the expectations.

- That is true, thanks for pointing it out! We adjusted the sentence accordingly: 

"Given the social restriction measures, a compromised sense of relatedness and connection should be expected; the findings pertaining to growing distances confirm this hypothesis and indicate that relatedness has become increasingly disrupted."

p.22 475 “For our study, the six interviewees participated four times throughout the study, adding up to 24 interviews in total.” contradicts p.11 224-226 “To ensure interrater reliability, two independent researchers executed this coding stage, one for the whole data set and the other for 8 of the 20 interviews.”

- Thanks for pointing out this mistake! It should indeed be 24 interviews at p.11, 24-226. We adjusted the text accordingly. 

p.23 “As most studies in this vein have been mainly qualitative,[46] researchers should pursue more quantitative insights, to generalize the results.” I cannot agree with this statement as there are numerous quantitative studies on this topic. You can find many of these papers on https://selfdeterminationtheory.org/topics/application-covid19/

- Again, thanks for pointing out additional quantitative studies regarding the topic. We omitted the sentence.

---

## [Decision Letter · Decision Letter 2]

4 Oct 2023

Caught between relief and unease: How university students’ well-being relates to their learning environment during the COVID-19 pandemic in the Netherlands

PONE-D-22-17942R2

Dear Dr. Kiltz,

We’re pleased to inform you that your manuscript has been judged scientifically suitable for publication and will be formally accepted for publication once it meets all outstanding technical requirements.

Kind regards,

Fatma Refaat Ahmed, Ph.D.

Academic Editor

PLOS ONE

Additional Editor Comments (optional):

Reviewers' comments:

Reviewer's Responses to Questions

**Comments to the Author**

1. If the authors have adequately addressed your comments raised in a previous round of review and you feel that this manuscript is now acceptable for publication, you may indicate that here to bypass the “Comments to the Author” section, enter your conflict of interest statement in the “Confidential to Editor” section, and submit your "Accept" recommendation.

Reviewer #3: All comments have been addressed

Reviewer #4: All comments have been addressed

2. Is the manuscript technically sound, and do the data support the conclusions?

Reviewer #3: Yes

Reviewer #4: Yes

3. Has the statistical analysis been performed appropriately and rigorously? 

Reviewer #3: Yes

Reviewer #4: N/A

4. Have the authors made all data underlying the findings in their manuscript fully available?

Reviewer #3: Yes

Reviewer #4: No

5. Is the manuscript presented in an intelligible fashion and written in standard English?

Reviewer #3: Yes

Reviewer #4: Yes

6. Review Comments to the Author

Reviewer #3: (No Response)

Reviewer #4: The authors have addressed the reviewer' comments in a good manner in this revision.

I would like to recommend this article for publication.

7. PLOS authors have the option to publish the peer review history of their article (what does this mean?). If published, this will include your full peer review and any attached files.

Reviewer #3: No

Reviewer #4: No

---

## [Editor Report · Acceptance letter]

24 Oct 2023

PONE-D-22-17942R2 

Caught between relief and unease: How university students’ well-being relates to their learning environment during the COVID-19 pandemic in the Netherlands 

Dear Dr. Kiltz:

I'm pleased to inform you that your manuscript has been deemed suitable for publication in PLOS ONE. Congratulations! Your manuscript is now with our production department. 

Kind regards, 

on behalf of

Dr. Fatma Refaat Ahmed 

Academic Editor

PLOS ONE